# Quantification of Cooking Method Effect on COP Content in Meat Types Using Triple Quadrupole GC-MS/MS

**DOI:** 10.3390/molecules25214978

**Published:** 2020-10-28

**Authors:** Shazamawati Zam Hashari, Alina Abdul Rahim, Goh Yong Meng, Suriya Kumari Ramiah

**Affiliations:** 1Food Biotechnology Program, Faculty of Science and Technology, University Sains Islam Malaysia, Bandar Baru Nilai 71800, Malaysia; shazazh@gmail.com (S.Z.H.); alina@usim.edu.my (A.A.R.); 2Department of Animal Production and Biodiversity, Institute of Tropical Agriculture and Food Security, University Putra Malaysia (UPM), Serdang 43400, Malaysia; ymgoh@upm.edu.my; 3Department of Veterinary Pre Clinical Science, Faculty of Veterinary Medicine, University Putra Malaysia, Persiaran UPM-Serdang, Serdang 43400, Malaysia

**Keywords:** cholesterol, cholesterol oxidation products, processed foods

## Abstract

A diet containing cholesterol is an essential component of biological function; however, cholesterol oxidation products (COPs) remain a major public health concern. This study investigated the effects of cooking methods (boiling and frying) on the production levels of COPs in processed foods. Samples, as represented by minced beef, chicken sausages, and fish fillets, were subjected to different cooking methods followed by COP extraction using a saponification method. Then, six common COPs, 5α-cholest, α-epoxy, β-epoxy, 25-HC, triol, and 7-keto, were quantified by triple quadrupole gas chromatography–mass spectrometry (GS-MS/MS). A significantly high number of COPs were detected in minced meat, of which 7-keto and triol were detected as major oxidation products, followed by chicken sausages and fish fillets (*p* ≤ 0.05). Compared to boiling, frying generated significantly more COPs, specifically triol (0.001–0.004 mg/kg) and 7-keto (0.001–0.200 mg/kg), in all samples. Interestingly, cholesterol level was found to be slightly (but not significantly) decreased in heat-treated samples due to oxidation during cooking, producing a higher number of COPs. Notably, the fish fillets were found to produce the fewest COPs due to the presence of a low amount of cholesterol and unsaturated fatty acids. In conclusion, adapting boiling as a way of cooking and choosing the right type of meat could serve to reduce COPs in processed foods.

## 1. Introduction

The consumption of meat provides nutritional value when appropriate methods have been applied to improve its shelf-life, texture, and taste and, importantly, to kill microorganisms [1,2,3]. In order to save time on meat preparation, processed meat products have gained popularity among a large portion of the population. To a greater extent, consuming precooked meat products is becoming more acceptable globally due to the lack of cooking time. Meats are rich in cholesterol and saturated fatty acids, which are considered unhealthy to frequently include in dietary habits [4]. Furthermore, a few processed meat foods (minced meat, sausages, ham, and bacon) are categorised as Group 1 carcinogens to humans by the International Agency for Research on Cancer [5].

Large, prospective US and European cohort studies and meta-analyses of epidemiological studies reported that long-term consumption of red meat, particularly processed meat, can lead to an increased risk of overall mortality, cardiovascular disease, colorectal cancer, and type 2 diabetes in both men and women [6]. The World Cancer Research Fund recommends not to eat more than 500 g (cooked weight) of red meat per week and to avoid processed meats to reduce cancer risk [7]. A meta-analysis study showed that consumption of unprocessed red meat (100 g/day) and processed meat (50 g/day) increased the risk of colorectal cancer by 17 percent and 18 percent, respectively [8]. In addition, red meat consumption may result in exposure to carcinogens through cooking methods, such as cooking meat at elevated temperatures through barbecue or smoking, or through nitrite preservation [9].

Cholesterol (5-cholesten-3β-ol) is an essential biological compound that is mainly distributed in foods of animal origin. Cholesterol is mainly involved in retaining the permeability and fluidity of the cell membrane [10]. Containing a double bond on carbon-5 makes it susceptible to oxidation. Cholesterol is usually present in the form of cholesterol esters, combined with long-chain fatty acids and in free form [11,12]. The degradation of cholesterol due to oxidation produces COPs. Small COP quantities are present in foods due to endogenous metabolic processes [13,14]. However, exogenous COPs can also be produced in foods due to many factors [13]. The presence of heat, radiation, light, free radicals, and metal ions can serve as sources of oxidation with different intensities [15]. In addition, cooking, illumination, and storage methods play a major role in meat oxidation, producing COPs [10,15,16,17,18]. Oxide products are similar to cholesterol in terms of structure, but they are dangerous to human health [19]. There are about 25 compounds classified as COPs, including 5,6α-epoxycholesterol, 5,6β-epoxycholesterol, 7α-hydroxycholesterol, 7-ketocholesterol, and 25-hydroxycholesterol [20]. These products have proven to be harmful to humans, causing undesirable effects on health status, such as triggering plaque formation in arteries. This leads to the development of coronary heart disease and atherosclerosis [12,20,21,22,23]. Besides that, COPs have proven to interfere with sterol metabolism and to be mutagenic, carcinogenic, atherogenic, and toxic to living cells. To a greater extent, COPs can alter properties of the cell membrane, eventually causing severe damage to the cells, interrupting cholesterol biosynthesis [24].

Processed foods are not usually eaten raw, but different types of heat treatment are necessary depending on individual choices. Direct heat generating high temperatures is extremely favourable for COP production. The large quantities of free radicals formed from high temperatures can catalyse cholesterol degradation at a faster rate [25]. A variety of cooking methods, especially frying, boiling, roasting, microwave heating, and air frying, are commonly practised in homes and in restaurants. Nevertheless, frying, followed by boiling, is the main method of choice for household cooking in Malaysia [26].

In order to identify the existence of COPs in foods, it is of great importance that the analytical methods are well validated. Therefore, proper method validation is crucial for the results to be considered reliable and accurate. However, the determination of COPs using triple quadrupole gas chromatography–mass spectrometry (GS-MS/MS) is still scarce. Most studies have found gas chromatography (GC) coupled with triple quadrupole tandem mass spectrometry (MS/MS) operated in electron ionisation mode (EI) to be an effective tool for (ultra)trace analysis [27]. Triple quadrupole MS/MS helps to establish the relationships between ions in a mass spectrum and identify the ions formed when the other ions in the spectrum undergo fragmentation [28]. According to Agilent Technologies [29], triple quadrupole mass spectrometry enables matrix interference to be drastically reduced, limiting the accuracy and detection of single ion monitoring (SIM) techniques.

The purpose of this research is to develop a method to analyse COPs in meat and poultry products using triple quadrupole gas chromatography–mass spectrometry (GC-MS/MS). Hence, this study investigates the effects of frying and boiling on COP production levels in processed meat products. For this purpose, locally produced chicken sausages, minced beef, and fish fillets were chosen for the experiment due to their popularity and because the effects of the cooking methods on COP formation remain unexplored. The concern over COP effects on consumers has led to the need for efficient COP extraction and the quantification of these compounds.

## 2. Materials and Methods

### 2.1. Reagents and Standards

The used standards were cholesterol, 5α-cholestane, 7-ketocholesterol, 5α,6α-epoxycholesterol, 5β,6β-epoxycholesterol, 25-hydroxycholesterol, and cholestane-3β,5α,6β-triol. All were obtained from Sigma Aldrich (St. Louis, MO, USA). N-trimethylsilylimidazole (TMSI) with pyridine (1:4) from Sulpeco were used as a derivatisation agent. All other chemicals, chromatography- and analytical-grade, were from Merck (Darmstadt, Germany).

### 2.2. Sample Preparation

Local-brand chicken sausages, minced beef, and fish fillets from local markets were bought and used as samples for this research. All samples were stored for one week in freezing temperature (−20 °C) under an inert (nitrogen) atmosphere before analysis was performed. On the day of analysis, samples were homogenised without being allowed to thaw and divided into three categories of treatment: raw (as reference or control samples), fried (using palm oil for about 10 min), and boiled (for about 10 min). Each raw and cooked sample was analysed in duplicate on the same day of treatment, and another replication of treatment and analysis was done at different times and on different batches of products. The total experiment was carried out in two batches, and analyses were done in duplicate for each batch (*n* = 8) for each product.

### 2.3. Instrumentation

Samples were analysed using triple quadrupole gas chromatography–mass spectrometry, Agilent 7000A GCMS-QQQ, using scan mode. Capillary column J&W 122-5532: DB-5Ms UI 30 m × 250 nm × 0.25 um was used to separate COPs.

### 2.4. Extraction Methods

The extraction method was based on that described by Bohac et al. [30], with some modifications during sample extraction. We accurately weighed 1 g of the samples to the nearest 0.0001 g into four 100 mL conical flasks. Samples were spiked with 0.00, 0.01, 0.02, and 0.05 mg of the standard. We added and homogenised 15 mL of chloroform:methanol (2:1, *v*/*v*). Then, 5 mL of distilled water was added, and the flask was shaken well. Each sample was centrifuged for 10 min at 2500 rpm. The upper layer was removed. The lower layer was kept dried until reaching a constant weight by putting the sample in a 70 °C water bath. Samples were saponified by adding 10 mL of 12% potassium hydroxide (KOH) in 90% ethanol. We then added 0.3 mL of freshly prepared pyrogallol solution (3%) as an antioxidant reagent. All conical flasks were covered with aluminium foil. The mixture was incubated at 80 °C for 15 min. Samples were allowed to cool to room temperature. Then, 5 mL of distilled water was added. The unsaponified matter was extracted with 2 times 10 mL hexane using a 100 mL separatory funnel. Both clear solutions at the upper layer were collected and combined. The extract was heated to dryness in a water bath at 45 °C. Samples were then added to 100 mL *N*-trimethylsilylimidazole (TMSI) with pyridine (1:4) and incubated in a water bath for 60 min at 60 °C. After incubation, the solvent in the samples was completely evaporated under a stream of nitrogen gas. Then, TMSI ether derivatives were dissolved in 1 mL of chromatography-grade hexane. Vortexing was used to make sure that the TMSI ether derivatives were completely dissolved in hexane. Dissolved samples were filtered using the Nylon syringe filter unit Millex^®^ (Merck, Darmstadt, Germany, 0.45 µm × 13 mm) from Millipore prior to GCMS-QQQ quantification in order to avoid damaging the capillary column and quadrupole system. All samples were quantified immediately after preparation.

### 2.5. Quantification Method

We injected 1 µL of each sample in splitless conditions with a 60 mL/min purge flow for 0.5 min. The injector temperature was 260 °C. The GC pressure was 21.997 psi with a septum purge flow of 3 mL/min. The gas saver was 20 mL/min after 2 min. The gas flow in the column was 1.2 mL/min, and the average velocity was 43.122 cm/s. The column temperature was set at 250 °C, increased to 280 °C at the rate 50 °C/min, and maintained at 280 °C for 5 min. The temperature was then increased to 300 °C at 50 °C/min and maintained for 6 min.

### 2.6. Statistical Analysis

The data were analysed using one-way analysis of variance (ANOVA) using Statistical Analysis System (SAS) to identify differences between means.

## 3. Results

Changes in the cholesterol and COP contents of minced beef, chicken sausages, and fish fillets varied following different cooking method (raw, boiled, and fried), as shown in Table 1. The content level of cholesterol was found to be decreased in all heat treatments for all sample types except for fried minced beef, in which it was increased by about 4%, and boiled fish fillets, where it increased by about 20%. Nonsignificant (*p* ≤ 0.05) changes in cholesterol content were noticed in boiled minced beef (3%), boiled chicken sausages (8%), fried chicken sausages (6%), and fried fish fillets (2%).

As indicated in Figure 1, the total COPs produced in fried samples were increased significantly (*p* ≤ 0.05) compared to boiling in all three samples. Frying consistently generated more COPs, especially triol (0.001–0.004 mg/kg) and 7-keto (0.001–0.200 mg/kg), compared to boiling in all the food samples. Among the tested samples, fried minced beef and fried fish fillets gave the highest amounts of 5α-cholestane (1.943 and 1.381 mg/kg) and 25-HC (0.004 and 0.004 mg/kg) compared to the raw and boiled treatments. As expected, the total COPs were increased after cooking, as a result of cholesterol oxidation, and different cooking methods which had various levels of impact on the production of different types of COPs. An interesting observation was made regarding the total COPs between the raw and cooked meat samples. High levels of total COPs (mg/kg) were encountered in raw minced beef and raw chicken sausages, which each had a similar range of total COPs as the fried samples. However, compared to raw and fried food samples, boiling yielded a lower level of total COPs, significantly in minced beef and nonsignificantly in chicken sausages.

## 4. Discussion

Cholesterol oxidation products (COPs) are considered more harmful than pure cholesterol to arterial cells and are directly linked to atherosclerosis and coronary heart disease [31]. Some of them are present in normal metabolic pathways in small quantities. However, excessive production of COPs has the potential to promote several harmful health effects, such as cytotoxic, mutagenic, carcinogenic, and atherogenic effects [32]. Increased levels of COPs (particularly 7-ketocholesterol and 7β-hydroxycholesterol) have been reported in disease states with increased oxidative stress, such as diabetes mellitus or combined family hyperlipidaemia [33]. Another study stated that hypertensive patients showed significantly higher COP concentrations (7-ketocholesterol, 5αcholestane-3β,5,6β-triol, and 5,6α-epoxy-5α-cholestan3α-ol) in serum than non-hypertensive ones [34].

The increased level of COPs in thermal processes might be caused by the oxidation of cholesterol by other substances during processing [16,17,35]. In order to avoid bias, the weight of dried food samples needed to be measured within a controlled environment to avoid the possibility of cholesterol oxidation, as the wet weight of raw and cooked samples may reduce cholesterol content [36]. COP levels in raw meat and chicken sausages are often detected due to the natural occurrence of enzymatic processes, and formation is directly related to cholesterol content [37]. Besides that, a high content of COPs in raw minced beef is due to a large contact surface area, which facilitates more oxidation [19]. In contrast, a different pattern was observed for fish fillets, where raw meat had the lowest number of total COPs, followed by boiled and fried fish fillets. These findings are concurrent with the statement of the United States Department of Agriculture (USDA) National Nutrient Database for Standard Reference Release 25, which reports that, in a 100 g sample, raw ground beef contains 71 mg of cholesterol, raw chicken sausages contain 66 mg, and fish fillets contain 37 mg, equivalent to 710, 660, and 370 mg/kg, respectively [38]. Therefore, a higher content of cholesterol in minced beef and chicken sausages is prone to oxidation when treated by heat. However, appreciable amounts of cholesterol in fish are prone to oxidation during the preparation and production of processed fish [39,40]. Many studies have shown that polyunsaturated fatty acids in fish are extremely susceptible to COP formation during fish processing [41,42]. Results can vary if different cooking methods are applied to all food samples. For example, when using microwave heating, the level of COP production was found to be higher [43]. However, the theory that higher cholesterol content generates a higher number of COPs is not applicable to all foods. A study by Galobart and Guardiola [44] reported that different foods with similar cholesterol contents may yield significantly different levels of cholesterol oxidation products [44]. For example, despite having a high content of cholesterol, only certain COPs are generated in eggs due to the physical attributes protecting the lipids from oxidation. Besides that, yolk containing cholesterol is protected by the shell from atmospheric oxygen, and the surrounding albumin is rich with antioxidants, providing further protection from oxidation [45,46]. In terms of cooking method, boiling remained the preferable method in all tested samples due to the low level of total COP production. In contrast, frying oil is often subjected to high temperatures and longer cooking times that can trigger the higher amount of COPs present in fried foods. The heating intensities of different cooking methods have variable destructive effects on meat structure that lead to differences in COP formation [47].

Oil usage is another crucial factor with regard to the high COP content in fried foods. The fatty acid composition of oil may be altered during frying, and oxidation takes place as a chemical reaction. As a result, oxide products may be absorbed into fried foods, especially meats [48]. As an alternative, vegetable oil containing antioxidant agents is advised to be used for deep-frying purposes [24]. The lowest COP concentration was found in triol and 7-keto for all the raw and boiled meat samples. However, this level rose and was recorded to be significantly higher in fried meat samples, suggesting that these COPs require higher temperatures and longer cooking times to be produced. Other studies have demonstrated that 7-keto abundantly forms during microwave heating as compared to frying [14]. Interestingly, a long heating time (>12 min) significantly elevated triol and 7-keto COPs [49,50]. Chicken sausages had the fewest of the other COPs (α-epoxy, β-epoxy, and 25-HC), which were nonsignificant in all heat- and nonheat-treated samples. A similar result was found by Alina et al [51], in which major COPs were not detectable at an earlier week of analysis [17]. However, a longer period of storage (approximately more than eight days of refrigerated storage) might be needed for significant detection. Therefore, it is advisable to avoid the longer storage of chicken sausages in order to minimise COP occurrence. The variation of different COP contents on differently treated food samples depends on multiple factors. The type of food packaging, storage conditions, and length and method of cooking time can alter the degree of COP generation [52,53]. In addition, the presence of activators, reactive oxygen species, and some enzymes on different types of processed meats can initiate cholesterol oxidation [32]. Interestingly, the presence of light can contribute to oxidation catalysed by natural pigments. Meat products that are stored in paper-based or transparent-film packaging are more prone to lipid oxidation due to longer exposure to light, which favours lipid oxidation, thus producing COPs at different intensities.

According to the Food and Agriculture Organization (FAO), the demand for edible vegetable oils on the international market has grown rapidly in recent years [54]. Palm oil is one of the most extensively consumed oils in the world. It has a balanced saturated–unsaturated fatty acid composition and good oxidative stability, which makes it the best choice for deep frying [55]. There is a growing interest in pomegranate seed oil as it is the major source of highly valued punicic acid (conjugated linolenic acid) in food systems [56]. Apart from that, extra virgin olive oil (EVOO), frequently used in Mediterranean diets either as a dressing or in cooking, has phenolic compounds with a strong antioxidant that promotes positive effects on human health [57]. Hence, choosing oil that contained a higher level of unsaturated fatty acids provided the best health benefit.

## 5. Conclusions

The long process of sample preparation for GC analysis may have contributed many errors in the form of systemic or random errors. Therefore, significantly simpler sample preparation processes could improve the GC analysis. Solid phase microextraction (SPME) can be performed to extract COPs prior to quantification by GCMS-QQQ to reduce errors and shorten the steps of analysis. There are no data to set the minimum limits for COPs in our daily food intake. Therefore, this study could raise important data to educate people and healthcare providers about food. In this study, we found that processed meat products subjected to heat treatment resulted in an increase of COP formation and a concurrent decrease in total cholesterol content. Frying significantly increased COP formation compared to boiling in the majority of the tested food samples. By comparison, minced beef contained a higher level of COPs than that of chicken sausages, and the raw and fried food samples had a similar range of total COPs. However, total COP content was found to be especially lower in fish fillets, both in the raw and boiled samples. Therefore, considering the type of processed food and desirable cooking methods is crucial; it can help minimise the oxidative degradation of cholesterol so that the safety and quality of food products can be assured.

## Figures and Tables

**Figure 1 molecules-25-04978-f001:**
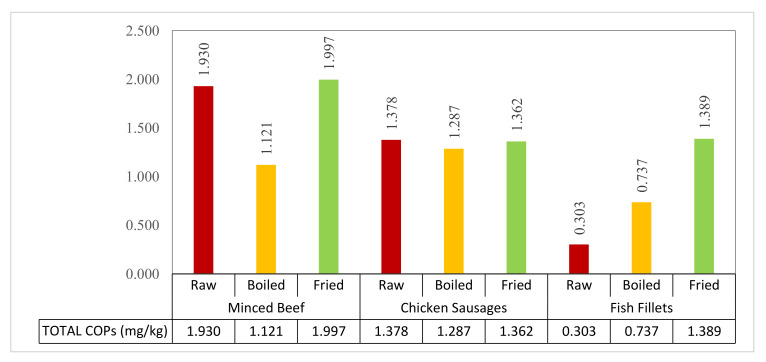
Total cholesterol oxidation product (COP) content (5α-cholest + α-epoxy + β-epoxy + 25-HC + triol + 7-keto; mg/kg) for three different treatment types (raw, boiled, and fried) of minced beef, chicken sausages, and fish fillets.

**Table 1 molecules-25-04978-t001:** Comparison of COP mean (ppm) in minced beef, chicken sausages, and fish fillets for three methods of treatment (raw, boiled, and fried).

Cooking Method	N	Cholesterol Oxidation Products (mg/kg)													
		5α-Cholest		Cholesterol		α-Epoxy		β-Epoxy		25-HC		Triol		7-Keto	
		Mean	SD	Mean	SD	Mean	SD	Mean	SD	Mean	SD	Mean	SD	Mean	SD
**Minced Beef**															
Reference	8	1.916 ^a^	0.159	395.227 ^a^	41.330	0.004 ^a^	0.001	0.004 ^b^	0.001	0.002 ^a^	0.020	0.000 ^b^	0.001	0.004 ^b^	0.001
Boiled	8	1.074 ^b^	0.364	382.521 ^a^	60.549	0.024 ^a^	0.007	0.019 ^a^	0.002	0.003 ^a^	0.001	0.001 ^b^	0.001	0.000 ^b^	0.000
Fried	8	1.943 ^a^	0.390	411.868 ^a^	65.268	0.022 ^a^	0.020	0.004 ^b^	0.002	0.004 ^a^	0.001	0.004 ^a^	0.001	0.020 ^a^	0.002
**Chicken Sausages**															
Reference	8	1.372 ^a^	0.382	170.727 ^a^	26.841	0.003 ^a^	0.001	0.000 ^a^	0.000	0.003 ^a^	0.001	0.000 ^a^	0.000	0.000 ^a^	0.000
Boiled	8	1.275 ^a^	0.181	157.083 ^a^	19.885	0.004 ^a^	0.001	0.003 ^a^	0.003	0.003 ^a^	0.002	0.002 ^a^	0.001	0.000 ^a^	0.000
Fried	8	1.329 ^a^	0.368	160.614 ^a^	38.845	0.020 ^a^	0.003	0.003 ^a^	0.003	0.002 ^a^	0.001	0.004 ^a^	0.001	0.004 ^a^	0.001
**Fish Fillets**															
Reference	8	0.297 ^c^	0.021	328.308 ^b^	62.457	0.000 ^a^	0.000	0.004 ^a^	0.004	0.002 ^a,b^	0.001	0.000 ^b^	0.000	0.000 ^a^	0.000
Boiled	8	0.734 ^b^	0.089	394.583 ^a^	35.726	0.000 ^a^	0.000	0.001 ^b^	0.001	0.001 ^b^	0.001	0.000 ^b^	0.000	0.001 ^a^	0.001
Fried	8	1.381 ^a^	0.476	323.114 ^b^	38.571	0.002 ^a^	0.001	0.000 ^b^	0.000	0.004 ^a^	0.004	0.001 ^a^	0.001	0.001 ^a^	0.001

Notes: COP = cholesterol oxidation product; ppm = parts per million; SD = standard deviation (*n* = 8); Reference = raw samples. ^a,b,c^ Means with the same letter in the same column for each type of product were not significantly different (α ≤ 0.05) based on Duncan’s multiple-range test.

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
