# Peer review of "Quantification of Cooking Method Effect on COP Content in Meat Types Using Triple Quadrupole GC-MS/MS"

_molecules, 2020, doi:10.3390/molecules25214978_

Round 1

Reviewer 1 Report

I’ve read with attention the paper of Kumari Ramiah et al., that is original and potentially of interest. The background and aim of the study have been clearly defined. The methodology applied is overall correct, the results are reliable and adequately discussed. I only would suggest the authors to shortly discuss the potential limitation of their research approach and their research perspective.

Reviewer 2 Report

Introduction

Please develop more the Introduction, focusing on the large consumption of  processed foods, from an epidemiological perspective and from a general health perspective

The aim/objectives of the study should be better and clearer described, in a separate paragraph, at the final of Introduction section. Which are the novelty aspects of this research?

Material and Methods

Please reshape "L73. Standards that were used were...", maybe: Used standards were

As the journal requests, Results and Discussion should be 2 separate sections. Please see the instructions for authors https://www.mdpi.com/journal/molecules/instructions 

I suggest to detail more the triple quadrupole gas chromatography mass spectrometry.

Also, I recommend to detail the health effects of the analysed molecules, the COPs, on different organs and cardiovascular system, in order to give a practical utility of this article. Also, discussion on the impact of the type of oil used in preparation of the foods would be beneficial.

Round 2

Reviewer 2 Report

The authors responded to all my requests. @ minor corrections:

Figure 1 must be introduced in section 3. Results., after L148.

L170. No need to define again COP. Please remove Cholesterol Oxidation Products. Please check the entire manuscript .

  • Abbreviations should be defined in parentheses the first time they appear in the abstract, main text, and in figure or table captions and used consistently thereafter.